# Accumulation of Fat Not Responsible for Femoral Head Necrosis, Revealed by Single-Cell RNA Sequencing: A Preliminary Study

**DOI:** 10.3390/biom13010171

**Published:** 2023-01-13

**Authors:** Yingjie Wang, Dandan Li, Haijia Chen, Zhuolin Li, Bin Feng, Xisheng Weng

**Affiliations:** 1Department of Orthopedic Surgery, State Key Laboratory of Complex Severe and Rare Diseases, Peking Union Medical College Hospital, Chinese Academy of Medical Science and Peking Union Medical College, Beijing 100730, China; 2Department of Laboratory Medicine, State Key Laboratory of Complex Severe and Rare Diseases, Peking Union Medical College Hospital, Chinese Academy of Medical Science and Peking Union Medical College, Beijing 100730, China; 3Guangzhou Saliai Stem Cell Science and Technology Co., Ltd., Guangzhou 510005, China

**Keywords:** osteonecrosis of femoral head, single-cell RNA-seq, cell types, substance metabolism, inflammatory reaction, blood vessel, bone metabolism

## Abstract

The etiology of osteonecrosis of the femoral head (ONFH) is not yet fully understood. However, ONFH is a common disease with high morbidity, and approximately one-third of cases are caused by glucocorticoids. We performed single-cell RNA sequencing of bone marrow to explore the effect of glucocorticoid on ONFH. Bone marrow samples of the proximal femur were extracted from four participants during total hip arthroplasty, including two participants diagnosed with ONFH for systemic lupus erythematosus (SLE) treated with glucocorticoids (the case group) and two participants with femoral neck fracture (the control group). Unbiased transcriptome-wide single-cell RNA sequencing analysis and computational analyses were performed. Seventeen molecularly defined cell types were identified in the studied samples, including significantly dysregulated neutrophils and B cells in the case group. Additionally, fatty acid synthesis and aerobic oxidation were repressed, while fatty acid beta-oxidation was enhanced. Our results also preliminarily clarified the roles of the inflammatory response, substance metabolism, vascular injury, angiogenesis, cell proliferation, apoptosis, and dysregulated coagulation and fibrinolysis in glucocorticoid-induced ONFH. Notably, we list the pathways that were markedly altered in glucocorticoid-induced ONFH with SLE compared with femoral head fracture, as well as their common genes, which are potential early therapeutic targets. Our results provide new insights into the mechanism of glucocorticoid-induced ONFH and present potential clues for effective and functional manipulation of human glucocorticoid-induced ONFH, which could improve patient outcomes.

## 1. Introduction

Osteonecrosis of the femoral head (ONFH) is a debilitating disease that mainly affects younger and active populations. It has been estimated that 100,000 to 200,000 patients are newly diagnosed each year in China [1]. Our previous epidemiological study found that the age (mean ± SD) of patients with ONFH was 46.45 ± 13.80 years, with a male predominance (70.1% of all ONFH cases). Of all patients with ONFH, 24.1% are steroid induced, among whom 18.53% are diagnosed with systemic lupus erythematosus (SLE), based on our previous study [2]. SLE is a systemic autoimmune disease that is clinically characterized by multi-system and multi-organ involvement [3]. The prevalence of SLE is estimated to be 30–50/100,000 worldwide [4,5,6], and 30–70/100,000 in the Chinese mainland [7,8]. Additionally, the prevalence of SLE has been gradually increasing, and increased from 64.99/100,000 in 1999 to 97.04/100,000 in 2012 in the United Kingdom [3]. However, SLE mortality has declined from 50% in the pre-corticosteroid era (circa 1948) to a 15-year survival of 85–95% in the modern era [4]. Glucocorticoids are commonly used in the treatment of SLE to induce remission and have been recommended by many international guidelines to control the disease [9,10,11,12]. It is worth noting that there is a close relationship between SLE and ONFH, and the incidence of ONFH is higher in patients with SLE than that in patients without SLE [13,14]. Additionally, it has been reported that the systemic use of glucocorticoids is an independent risk factor in ONFH, and the incidence of ONFH has been shown to be 6.7% with glucocorticoids treatment > 2.0 g [15,16,17]. Although hip arthroplasty has been shown to be successful in curing end-stage ONFH, it will bring a heavy economic burden to patients and society. Accordingly, it is of great clinical significance to investigate the underlying mechanism of glucocorticoid-induced ONFH to facilitate the development of earlier and less invasive therapy to prevent and even reverse the process of ONFH and further hip arthroplasty. 

Most recent studies concerning glucocorticoid-induced ONFH focus on the osteogenesis and angiogenesis and adipogenesis potential and apoptosis of osteogenic cells [18,19,20], and few researchers have systematically studied the alternations of other cell types in the microenvironment surrounding osteogenic cells in response to glucocorticoid therapy. Our previous study showed that a high dosage of glucocorticoids promoted adipogenic differentiation and cell apoptosis, and inhibited the osteogenic differentiation and cell proliferation of hBMSCs [21], which has also been demonstrated by other research groups [22,23]. To further illustrate the underlying molecular mechanism, our team found that long non-coding RNA (lncRNA) RP11-154D6 promoted osteogenic differentiation while inhibiting adipogenic differentiation of hBMSCs [24]. Recently, the changes of cell types and biological functions in the context of SLE have attracted increasing attention based on the results of single-cell RNA sequencing analysis (scRNA-seq). Previous studies have reported a high interferon-stimulated gene expression signature derived from a small number of transcriptionally defined subpopulations within major cell types in peripheral blood mononuclear cells, including monocytes, CD4+ and CD8+ T cells, natural killer cells, conventional and plasmacytoid dendritic cells, B cells, and especially plasma cells [25]. Compared to other immune cell subsets, low-density granulocytes display the highest expression of interferon-inducible genes. Additionally, low-density granulocytes in peripheral blood of SLE patients increase and may promote immune dysregulation and prominent vascular damage characteristic of the disease [26]. Studies have shown that long-term use of glucocorticoids may lead to the following changes, including but not limited to abnormal lipid metabolism [27,28], vascular endothelial cell damage [27,29], micro-thrombosis [27,30], and high intraosseous pressure of the femoral head [31], all of which eventually cause ONFH. Interestingly, the relationship among these changes needs to be further clarified, and the corresponding cell types with changed biological functions in bone marrow microenvironment remain unclear. 

In this study, bone marrow samples from the proximal femur were extracted from two patients with ONFH for SLE and two with femoral neck fracture during hip arthroplasty. We compared the overall differences in cell types and biological functions and their relationships in bone marrow using scRNA-seq analysis. To the best of our knowledge, this is the first study to systematically explore the changes in cell types and biological functions of bone marrow in glucocorticoid-induced ONFH and their relationships. Our results may provide targets, such as specific cell types or genes, for early treatment for glucocorticoid-induced ONFH in SLE patients to prevent possible invasive operations. 

## 2. Materials and Methods

### 2.1. Obtaining Bone Marrow Samples of Femoral Head Necrosis

Bone marrow of the proximal femur was extracted from two male participants (39 and 71 years old) diagnosed with femoral neck fracture and two female participants (19 and 32 years old) diagnosed with ONFH after glucocorticoid therapy for SLE during hip arthroplasty. The two SLE patients were diagnosed with ONFH by magnetic resonance imaging 1 year and 5 years, respectively, after methylprednisolone therapy, and the cumulative dosage of methylprednisolone was more than 4 g and 15 g, respectively. The total amount of extracted bone marrow from each participant was at least 10 mL. 

This study was approved by the Ethics Committee of Peking Union Medical College Hospital (No. JS-2447) and was conducted in accordance with the International Conference on Harmonization Good Clinical Practice guideline, the Declaration of Helsinki, and applicable regulatory requirements. All patients provided written informed consent prior to participation in the study.

### 2.2. scRNA-seq Data Alignment and Sample Aggregation

Raw sequencing data (bel files) were converted to fastq files using Ilumina bcl2fasta. version 2.19.1 and were aligned to the human genome reference sequence (http://cf.10xgenomics.com/supp/cell-exp/refdata-cellranger-GRCh38-1.2.0.tar.gz (accessed on 8 March 2022)).

The CellRanger (10X Genomics) analysis pipeline was used to generate a digital gene expression matrix from the data. The raw digital gene expression matrix (UMI counts per gene per cell) was filtered, integrated, normalized, and clustered using the R package Seurat v3 (https://satijalab.org/ (accessed on 8 March 2022)) as follows: cells with a very small library size (<500 UMI counts and <500 genes detected) and a very high (>0.2) mitochondrial genome transcript ratio were removed. Genes that were detected (UMI count > 0) in less than three cells were removed. As an unusually high number of genes can result from a “doublet” event, in which two different cell types are captured together with the same barcoded bead, cells with >6000 genes were also removed. The filtered gene expression matrix of all four samples was integrated with the Seurat v3. In brief, we identified 2000 features with high cell-to-cell variation, as described above. Next, we identified “anchors” between individual datasets with the FindIntegrationAnchors function and inputted these anchors into the IntegrateData function to create a “batch-corrected” expression matrix of all cells, which allowed cells from different datasets to be integrated and analyzed together.

### 2.3. Dimensionality Reduction and Clustering Analysis

The gene expression values were then subjected to library size normalization, in which raw gene counts from each cell were normalized relative to the total number of read counts present in that cell. The resulting expression values were then multiplied by 10,000 and log-transformed. Subsequent analyses were conducted using only the most highly variable genes in the dataset. Principal component analysis (PCA) was used for dimensionality reduction, followed by clustering in the PCA space using a graph-based clustering approach. T-distributed stochastic neighbor embedding (t-SNE) was then used for two-dimensional visualization of the resulting clusters.

### 2.4. Cell Type Annotation and Cluster Marker Identification

The FindAllMarkers function in Seurat was used to identify markers for each of the identified clusters. Clusters were then classified and annotated based on the expression of canonical markers of cell types. Clusters that expressed two or more canonical cell type markers were classified as doublet cells, and clusters that expressed no canonical cell type markers were classified as low-quality cells. Doublet cell clusters and low-quality cell clusters were excluded from further analyses. 

### 2.5. Gene Functional Annotation

Gene ontology (GO), gene-set enrichment analysis, and Kyoto Encyclopedia of Genes and Genomes (KEGG) pathway analyses of differentially expressed genes (DEGs) were performed using gene set variation analysis (GSVA) and clusterProfiler, which supports statistical analysis and visualization of functional profiles for genes and gene clusters. 

## 3. Results

### 3.1. Single-Cell Profiling of Human Bone Marrow

To define the gene expression patterns of various cell types in the bone marrow of glucocorticoid induced femoral head necrosis, four bone marrow samples were obtained from the proximal femur of four participants undergoing total hip arthroplasty. Of the four participants, two were diagnosed with glucocorticoid-induced ONFH, which was applied to control the condition of SLE (the case group), and the other two with femoral neck fracture (the control group). In total, 7080 and 10,325 individual cells in the case and control groups, respectively, were profiled by droplet-based scRNA-seq after rigorous filtration (Figure 1A). To identify the transcriptional states, we analyzed the cellular compositions of human bone marrow using uniform manifold approximation and projection (UMAP) [32,33,34,35]. Unbiased clustering of all cells in both groups resulted in 25 clusters (Figure 1B), spanning progenitor, T cell, B cell, natural killer cell, granulocyte, mononuclear macrophage, erythroblast, and bone marrow cell. In particular, 17 cell types were specified by marker genes: (1) myelocyte (cluster 0′ and 1′ and 24′, highly expressing *Camp* and *Ltf* and *Lcn2* and *Mmp8*); (2) neutrophil (cluster 2′ and 3′, highly expressing *G0s2* and *Cmtm2* and *IL1R2* and *Rgs2*); (3) pro-myelocyte (cluster 4′ and 7′ and 10′, highly expressing *Defa4* and *Azu1* and *Elane* and *Mpo*); (4) T cell: CD4+ central memory (cluster 5′, highly expressing *IL7R* and *CD3d* and *IL32* and *CD3e*); (5) T cell: CD8+ (cluster 6′, highly expressing *CCL5* and *Gzma* and *Gzmk* and *Klrd1*); (6) B cell: immature (cluster 8′ and 12′, highly expressing *CD74* and *Ighm* and *CD79b* and *Ms4a1*); (7) monocyte: CD16- (cluster 9′ and 16′, highly expressing *Vcan* and *CD14* and *Aif1* and *Klf4*); (8) T cell: CD4+ (cluster 11′, highly expressing *IL7R* and *Ccr7* and *Trbc1*); (9) B cell: plasma cell (cluster 13′, highly expressing *Tnfrsf17* and *Sdc1* and *Fcrl5* and *Mzb1*); (10) granulocyte macrophage progenitor (GMP) (cluster 14′, highly expressing *Gstp1* and *Igfbp7* and *Mgst1* and *Ran*); (11) common myeloid progenitor (CMP; cluster 15′, highly expressing *Avp* and *Fam69b* and *Cytl1* and *Cpxm1*); (12) erythroblast (cluster 17′, highly expressing *Ackr1* and *Pklr* and *Klf1* and *Rhag*); (13) pro-B cell: CD34+ (cluster 18′ and 19′, highly expressing *Troap* and *Plk1* and *Vpreb3* and *Mad1l1*); (14) bone marrow (BM) cell (cluster 20′, highly expressing *Tspo2* and *Cr1l* and *Slc4a1*); (15) natural killer (NK) cell (cluster 21′, highly expressing *Klrf1* and *Prf1* and *Sh2d1b* and *CD247*); (16) pre-B cell: CD34− (cluster 22′, highly expressing *Lamp5* and *Irf7*); and (17) monocyte-derived macrophage: IL-4/Dex/cntrl (cluster 23′, highly expressing *Mrc1* and *Vsig4* and *Apoe* and *Pla2g7*) (Figure 1C,D, Appendix A). To help resolve differentiation relations among cells, we used correlations of average expression profiles between clusters (Figure 1E). Figure 2A,B show that the cell types were the same in both groups, but the number of the cell types was various. In general, the total cell number in the case group (7080 cells) was less than that in the control group (10,325 cells). Compared to the control group, the number of neutrophil, myelocyte, pro-myelocyte, monocyte-derived macrophage: IL-4/Dex/cntrl, monocyte: CD16−, pre-B cell: CD34−, and NK cell decreased, while the number of pro-B cell: CD34+, immature B cell, plasma B cell, and BM cell increased (Figure 2C–E). 

### 3.2. Expression of Glucocorticoid Therapy-Related Genes in Various Cell Types

To reveal the distribution of ONFH-related genes in cell types, the gene list was extracted from the Comparative Toxicogenomics Database; these genes intersect with the results of scRNA-seq, which demonstrates that almost all cell types are closely related to ONFH (Figure 3A). Additionally, 65 genes related to glucocorticoid therapy were obtained from the intersection of the Comparative Toxicogenomics Database and Gene Set Enrichment Analysis Database (Appendix A), which then intersected with the results of scRNA-seq. Figure 3B shows that glucocorticoid therapy-related genes are also expressed in almost all cell types. Of all glucocorticoid therapy-related genes, 16 upregulated genes were mainly expressed in neutrophil, myelocyte, pro-myelocyte, BM cell, and CD16− monocyte (Appendix A), while 49 downregulated genes were mainly expressed in pro-myelocyte, pro-B cell (CD34+), GMP, CMP, and erythroblast (Appendix A). GO and KEGG analyses were performed for 65 genes associated with glucocorticoid therapy. Initially, the affected biological processes (BP) included neutrophil degranulation, response to oxidative stress, hydrogen peroxide metabolic process, reactive oxygen species metabolic process, positive regulation of neutrophil migration, and cell redox homeostasis (Figure 3C). Cellular component (CC) analysis revealed changes in the mitochondrial matrix, mitochondrial nucleoid, pore complex, lipid droplet, and others (Figure 3D). Additionally, molecular function (MF) analysis revealed peroxiredoxin activity, oxidoreductase activity acting on peroxide as an acceptor, thioredoxin peroxidase activity, antioxidant activity, nitric oxide synthase regulator activity, peroxidase activity, kinase regulator activity, and others (Figure 3E). Finally, the enriched KEGG signaling pathways covered necroptosis, NOD-like receptor, IL-17, and NF-kappa B (NF-κB) signaling pathways (Figure 3F). Both the results of GO and KEGG were primarily related to the inflammatory response, cell proliferation and apoptosis, substance metabolism, and vascular function. 

### 3.3. Pathways and Corresponding Key Genes in Various Cell Types

Next, Gene Set Enrichment Analysis (GSEA) and Gene Set Variation Analysis (GSVA) were performed to reveal the changes in interacted signaling pathways in various cell types [36,37,38]. Overall, the affected 28 pathways associated with glucocorticoid-induced ONFH included calcium, insulin, P53, cell cycle, apoptosis, mTOR, Wnt, MAPK, NOD and Toll-like receptor, glycolysis and gluconeogenesis, pyruvate metabolism, oxidative phosphorylation, pentose phosphate, lysosome, peroxisome, glutathione metabolism, fatty acid metabolism, peroxisome proliferator-activated receptor (PPAR), citrate cycle TCA cycle, adipocytokine, JAK-STAT, complement and coagulation cascades, leukocyte trans-endothelial migration, vascular smooth muscle contraction, systemic lupus erythematosus, TGF-β signaling pathways, NOTCH signaling pathway, and others (Table 1). It is noteworthy that the above pathways which are most relevant to the inflammatory response, glucose and lipid metabolism, vascular injury and angiogenesis, dysregulated coagulation and fibrinolysis, cell proliferation and apoptosis, and osteogenesis are basically consistent with the results of GO and KEGG of 65 genes affected by glucocorticoid. Of the mentioned 28 pathways, 78 DEGs occurred simultaneously in multiple pathways, indicating that 78 DEGs connect 28 pathways. For instance, P53 and cell cycle signaling pathways share 13 DEGs; TNF plays a role in angiogenesis by inducing vascular endothelial growth factor (VEGF) production synergistically with IL-1B and IL-6 [39] and is the network hub of MAPK, TLR, NLR, apoptosis, SLE, TGF-beta, and adipocytokine signaling pathways. 

The distribution of 78 DEGs in signaling pathways and the number of signaling pathways affected by an individual gene are shown in Figure 4. A protein–protein interaction (PPI) network corresponding to 78 DEG-encoding proteins was constructed to demonstrate the relationship of the multiple signaling pathways mentioned above (Figure 5).

### 3.4. Changes in Metabolic Profiling in the Bone Marrow

According to previous studies, abnormal lipid metabolism had been reported to play a significant role in ONFH [27,28], while glucose, lipid, and protein substance metabolism related closely. Based on the results of the current study, there are three major substance metabolism disorders in various cell types of bone marrow. 

The results of GSVA and GSEA indicated that the substance metabolism of erythroblasts changed the most. GSVA revealed that the regulated pathways in the case group included down-regulated peroxisome, oxidative phosphorylation, fatty acid metabolism, pyruvate metabolism, glycolysis gluconeogenesis, Wnt pathways, and up-regulated PPAR and TGF-beta pathways (Figure 6A). Additionally, GSEA demonstrated up-regulated adipocytokine (Figure 6B), P53 signaling pathways (Figure 6C), leukocyte trans-endothelial migration (Figure 6D), as well as down-regulated citrate cycle TCA cycle (Figure 6E), glutathione metabolism (Figure 6F), and cell cycle (Figure 6G). 

Initially, the number of erythroblasts in the case group (145 cells) was less than that in the control group (165 cells) (Figure 2C–E), which is consistent with the phenomenon that anemia often occurs in patients with SLE [40,41]. Furthermore, the mostly affected signaling pathways of erythroblast mainly involved substance metabolism of glucose and fat. The protein–protein interaction (PPI) network of the 29 DEGs expressed in erythroblast is shown in Appendix A. Of the 29 DEGs, the low expression of *Myc* in the case group compared with the control group (Appendix A) connects substance metabolism and cell proliferation-related pathways (Appendix A). Above all, low expression of DEGs concerning glucose-associated aerobic oxidation was detected, including *Pkm*, *Pklr*, *Ldha*, *Ldhb*, *Idh1*, *Idh2*, *Sdha*, and *Phda1*. Additionally, mitochondrial fatty acid beta-oxidation pathway associated DEGs, including *Acat1* and *Acsl6*, were expressed at higher levels in the case group than in the control group. Besides fatty acid beta-oxidation, *Mdh1* and *Mdh2*, which are involved in fatty acid synthesis, were expressed at lower levels in the case group than in the control group. Finally, highly expressed DEGs (*Gadd45b*, *Gadd45g*, *Ccnb1*, *Ccnb2*, *Cdk1*, *Cdkn1a*, *Ccne2*, and *E2F4*) and lowly expressed DEGs (*Ccne1* and *Cdk6*), both of which play a role in the G1/S/G2 transition, together caused a slight decrease in the number of erythroblasts in the case group. The metabolic changes in other cell types were consistent with those observed in erythroblast. 

### 3.5. Changes in Inflammatory-Response Profiling in Bone Marrow

Monocyte-derived macrophages actively contribute to inflammation [42]. Our study demonstrated that the number of monocyte-derived macrophage in the case group was 0.26 times that of the control group (24 versus 92 cells). In addition to total quantity, the number of activated macrophages in the case group was less than that in the control group, as indicated by the level of IL-1B (secreted by activated macrophages). 

The results of GSVA and GSEA indicated that the inflammation-related pathways of monocyte-derived macrophage changed the most. The most significantly affected inflammation-related pathways in the case group were down-regulated MAPK, NLR, peroxisome, and systemic lupus erythematosus, and up-regulated leukocyte trans-endothelial migration, Wnt, JAK-STAT, and complement and coagulation cascades pathways (Figure 6H–K), all of which are associated with SLE, inflammatory response, blood vessel development, and glucocorticoid therapy. *Jun*, one of the targeted genes of glucocorticoid therapy, serves as the network hub and promotes steroidogenic gene expression [43], while the JAK-STAT and MAPK pathways can be considered as the center of the inflammatory response (Appendix A). The expression level of IL-6 in the case group was higher than that in the control group, as revealed by GSEA analysis (Figure 7A). Additionally, the upregulation of IL-6 was associated with high expression of MAPK11 (Figure 7B) [44,45]. However, the level of IL-10, which has profound anti-inflammatory functions on many immune cells, was downregulated in the case group (Figure 7C), and was probably responsible for glucocorticoid therapy. Meanwhile, SOCS3, the negative regulator of JAK-STAT [46,47], decreased significantly (Figure 7D). 

In addition to the JKA-STAT pathway, the MAPK pathway also plays a significant role in the inflammatory response of macrophages [48,49,50,51]. It was noted that MAP2K2, the negative regulator of the MAPK pathway [52,53], was highly expressed in the case group (Figure 7E) and the level of IL-1B decreased (Figure 7F). IL-1B is a potent proinflammatory cytokine that induces prostaglandin synthesis, neutrophil influx and activation, T-cell activation and cytokine production, B-cell activation and antibody production, fibroblast proliferation, and collagen production [54]. Interestingly, complement activation associated with C1qA and C1r was highly expressed in the case group (Figure 7G,H), and the complement and coagulation cascades were also activated in the case group (Figure 6K). With regard to blood vessels, *Myc* was downregulated in the case group. 

### 3.6. Potential Effects of SLE Combined with Glucocorticoids on Blood Vessel and Bone Metabolism

With regard to the effects of SLE combined with glucocorticoids on blood vessel and bone metabolism, our study showed that the changes in immature B cell signaling pathways of angiogenesis and osteogenesis may be the most representative. The number of immature B cells in the case group was 3.86 times that in the control group (Figure 2C). The altered pathways relevant to blood vessel and bone metabolism were as follows: (1) activated calcium and vascular smooth muscle contraction pathways; (2) inhibited leukocyte trans-endothelial migration, SLE, complement and coagulation cascades, and MAPK pathways (Figure 6L) [55,56]. 

*Jun*, a glucocorticoid-targeted gene, serves as a network hub, which is constructed by multiple affected pathways that link anti-inflammatory, pro-inflammatory, bone metabolism, and even cell proliferation-related signaling pathways (Appendix A). Based on the number and function of the involved genes, MAPK and vascular smooth muscle contraction are probably located at the network hub regulating the structure and function of blood vessels. The MAPK pathway was found to be repressed, and the vascular smooth muscle contraction pathway was activated in the case group (Figure 6L). Additionally, the levels of CD40 and MAP3K8 (Figure 7I,J), which are positive regulators of MAPK, were downregulated in the case group. Moreover, the case group showed high expression of GADD45B and GADD45G (Figure 7K,L), which activate MTK1/MEKK4 kinase to activate both p38 and JNK MAPK. Meanwhile, the expression of nuclear factor-kappa B inhibitor alpha (NFKBIA) (Figure 7M), a member of the nuclear factor-kappa B (NFKB) inhibitor family, was increased in the case group, and the expression of CD14 (Figure 7N), leading to NF-κB activation and cytokine secretion and inflammatory response [57], was decreased in the case group. In the vascular smooth muscle contraction pathway, MYLK (Figure 7O), which triggers endothelial contraction during the development of microvascular hyperpermeability by phosphorylating myosin light chains, was upregulated, while Myc (Appendix A), which promotes VEGFA production and subsequent sprouting angiogenesis, was downregulated [58]. Furthermore, CTSK is involved in osteoclastic bone resorption and may participate partially in the disorder of bone remodeling, which displays potent endoprotease activity against fibrinogen at acid pH and may play an important role in extracellular matrix degradation [59]. CTSK was also highly expressed in the case group, and the lysosome pathway in the case group was inhibited compared to the control group (Figure 6M).

### 3.7. Potential Effects of SLE Combined with Glucocorticoids on Cell Proliferation and Apoptosis

Figure 2C shows fewer neutrophils in the case group than in the control group (1101 versus 2143 cells). GSVA demonstrated activation of the Wnt pathway, and inhibition of insulin, P53, calcium pathways, and the glycolysis gluconeogenesis pathway in case group (Figure 6N). *Ccnd1* served as the network hub of multiple pathways and was highly expressed in the case group (Appendix A). Increased GTSE1 binds to the tumor suppressor protein p53 to prevent apoptosis [60,61]. Besides GTSE1, other genes are also known to participate in cell apoptosis: For instance, (1) increased VDAC1 may participate in the formation of the permeability transition pore complex (PTPC), which is responsible for the release of mitochondrial products that trigger apoptosis [62,63]; (2) decreased EIF4EBP1 and RRM2 facilitate cell proliferation [64,65]; (3) increased CCND1 forming a complex with, and functions as a regulatory subunit of, CDK4 or CDK6, whose activity is required for cell cycle G1/S transition (CDK6 expression was relatively low in the case group) [66,67]; and (4) low expression of CHEK1 and CDKN1A serve to inhibit cell proliferation [68,69,70]. 

Enolase 1 (ENO1) is downregulated in case group (Appendix A), which promotes the cell proliferation [71,72]. Besides ENO1, thrombospondin 1 (THBS1), the ligand for CD36, which has antiangiogenic properties and cell-to-cell/matrix interactions [73,74], was higher in the case group (Appendix A).

## 4. Discussion

The prevalence of SLE is gradually increasing, and SLE is an independent risk factor for ONFH [3,75]. Glucocorticoids were first introduced for the treatment of SLE in 1948 and are one of the main drugs recommended by numerous clinical guidelines [10,11,12]. Glucocorticoid induced osteonecrosis was first observed as early as 1960 [9,10,11,12]. 

Although numerous studies have been performed in recent years to elucidate the pathogenesis of glucocorticoid-induced ONFH, the underlying mechanisms remain largely unknown. Many theories have been proposed to explain glucocorticoid-induced ONFH, among which the representative perspectives include intraosseous high pressure, change in coagulation mechanism, fat embolism, osteoporosis, and osteocyte apoptosis theory [27,76]. Additionally, studies have shown that long-term and high-dose systematic use of glucocorticoids may lead to abnormal lipid metabolism [77], vascular endothelial cell damage [78], microthrombosis [79], and high pressure in the femoral head [80], ultimately causing glucocorticoid-induced ONFH. In recent years, an increasing number of studies have focused on hBMSCs, which has led to an increase in the number of key genes confirmed to be associated with ONFH. 

Studies have shown that long-term application of glucocorticoids can affect the coagulation pathway by changing the expression of plasminogen activator inhibitor-1 (PAI-1) and alpha-2 macroglobulin (A2M), both of which lead to the formation of microthrombosis and ONFH [81,82]. In addition, the downregulation of hypoxia-inducible factor-1α (HIF-1α) and VEGF gene expression leads to a reduction in angiogenesis, which also plays an important role in ONFH [83]. Finally, disordered lipid and cholesterol metabolism may also increase the risk of ONFH [84,85]. Furthermore, it is also significant to systematically clarify the changes in the biological properties of other cell types surrounding hBMSC. Additionally, changes in biological processes of other cell types and their interaction mechanisms remain to be elucidated.

Overall, the present study demonstrates the changes in single-cell profiling of human bone marrow and changes in the biological processes of various cell types, mainly associated with the inflammatory response, substance metabolism, dysregulated coagulation and fibrinolysis, vascular injury and angiogenesis, and cell proliferation and apoptosis. 

SLE is a chronic diffuse connective tissue disease with unclear etiology. SLE involves multiple organs and tissues and often leads to abnormalities in the blood system. Indeed, it has been reported that neutrophil dysregulation is implicated in the pathogenesis of SLE. Low-density granulocytes were previously identified as a proinflammatory neutrophil subset that differs functionally from autologous lupus normal density neutrophils and healthy control neutrophils. Low-density granulocytes induce endothelial damage and vascular dysfunction in vitro, through their enhanced ability to synthesize and extrude neutrophil extracellular traps [26]. The present study found that the number of neutrophils, myelocytes, and pro-myelocytes in patients with SLE is 0.51, 0.39, and 0.30 times that observed in patients with femoral neck fracture. Therefore, the presence and proportion of low-density granulocytes in bone marrow of ONFH patient with SLE warrant further study. Furthermore, the production of pre-B cells (CD34−), monocytes (CD16−), and monocyte-derived macrophages decreased in case group compared with that in control group. Moreover, bone marrow abnormalities, such as myelofibrosis, pure red cell aplasia, aplastic anemia, and features suggestive of myelodysplastic syndromes, have been reported in patients with SLE, suggesting that the bone marrow may also be a target organ in the disease [86,87]. Although a previous study suggested that the dysregulation of B cell development within the peripheral blood is involved in SLE, there is a lack of evidence concerning the specific condition of B cell development in the bone marrow in this patient group [88]. The present study shows decreased pre-B cells (CD34−). Furthermore, Kubo et al. reported that the proportion of B cell plasma within peripheral blood was higher in SLE patients and correlated with disease activity [89,90]. The results of our study further confirmed that not only plasma B cells, but also pro-B cells (CD34−) and immature B cells were increased in case group. Interestingly, the production of pre-B cells (CD34−) decreased, while the production of pro-B cells (CD34−), immature B cells, and plasma B cells increased. This phenomenon may be related to the decrease in self-renewal and differentiation ability of bone marrow hematopoietic progenitor cells. The number of monocytes (CD16−) and monocyte-derived macrophages decreased. 

Based on the Comparative Toxicogenomics Database and Gene Set Enrichment Analysis Database, 65 glucocorticoid-associated genes were selected (Appendix A). The 65 dysregulated genes were mainly expressed in neutrophil, myelocyte, pro-myelocyte, BM cells, and monocytes (CD16−) (Appendix A), while the downregulated genes were mainly expressed in pro-myelocyte, pro-B cell (CD34+), GMP, CMP, and erythroblast (Appendix A). Clinical studies in patients with anemia and polycythemia have suggested a functional link between erythropoiesis and bone homeostasis. Both anemic and polycythemic patients are at risk for bone disease [91]. Hematopoiesis, erythropoiesis, bone mineralization, and bone marrow homeostasis are all important macrophage-dependent processes that occur within the bone marrow microenvironment [92,93,94,95]. 

Based on the results of the present study and a review of the available literature, the biological properties of erythroblast, monocyte-derived macrophage, immature B cell, and neutrophil were discussed. For erythroblast, the most noticeable changes were within the metabolism of glucose and fat. Glucocorticoids induce an increase in lipolysis, lipid mobilization, liponeogenesis, and adipogenesis, and increased triglyceride levels are also typically observed [96,97]. The processes of aerobic oxidation of glucose and fatty acid synthesis are repressed, while the mitochondrial fatty acid beta-oxidation pathway is activated. The changes in glucose and fat metabolism in erythroblast may be an adaptive change to reduce the level of blood fat. According to our previous study, the adipogenic differentiation of hBMSCs was enhanced, and hypertrophic adipocytes led to high pressure in the femoral head and subsequent femoral head necrosis [21,24]. Taken together, it could be speculated that lipid droplets may only be present in adipocytes. 

Activated JAK-STAT and repressed MAPK pathways are observed in macrophage. Blocking the JAK-STAT signaling pathway has been reported to reduce the secretion of inflammatory cytokines and cartilage matrix-degrading enzymes and markedly alleviates degradation of the cartilage matrix. Furthermore, activation of MAPK may help to prevent osteoporosis [98,99]. In a previously published study [100], the levels of IL-6 and IL-10 in patients with SLE increased significantly and were positively correlated with disease activity. Upregulation of IL-6 and IL-6R can activate the JAK-STAT1 or STAT3 signaling pathway for signal transduction, which is induced by high expression of MAPK11. Additionally, IL-10, a negative regulator of the JAK-STAT and NF-KB pathways [101], is downregulated in case group, which targets antigen-presenting cells (APCs), such as macrophages and monocytes, and inhibits their release of pro-inflammatory cytokines, including granulocyte-macrophage colony-stimulating factor (GM-CSF), granulocyte colony-stimulating factor (G-CSF), IL-1 alpha, IL-1 beta, IL-6, IL-8, and TNF-alpha [102,103,104]. The decreased expression of IL-10 indicates that the effects of SLE on IL-10 and JAK-STAT maybe more pronounced than those of glucocorticoids, which may be related to the fact that IL-10 is not a direct target gene of glucocorticoids. 

*Jun* is a potential therapeutic target to reduce the incidence of glucocorticoid-induced ONFH for the following reasons: (1) *Jun* is one of the target genes of glucocorticoid therapy to promote steroidogenic gene expression [43] and (2) *Jun* serves as the network hub of the inflammatory response (Appendix A). 

It had been reported that an immature B cell population can serve as a marker for monitoring tumor angiogenesis and anti-angiogenesis therapy in mouse models [105]. Additionally, the dysregulation of immature B cell is associated with bone loss [106]. For immature B cell, activated calcium and vascular smooth muscle contraction and suppressed MAPK pathways influence vascular function and angiogenesis. Moreover, upregulated MYLK, a common gene of calcium and vascular smooth muscle contraction pathways, triggers endothelial contraction. Myc is downregulated, which can activate the transcription of growth-related genes and bind to the VEGFA promoter to facilitate VEGFA production and subsequent sprouting angiogenesis [58] and platelet-derived growth factor receptor beta (PDGFRB), which is essential for normal blood vessel development and repair of vascular injury sites [107]. Additionally, complement and coagulation cascade pathways activated by immune complexes were suppressed in patients with SLE treated by glucocorticoids, indicating that activated complement does not damage the blood vessels, which may be responsible for the immunosuppressive effects of glucocorticoids. 

CTSK is predominantly expressed in osteoclasts, mutations of which lead to pycnodysostosis [108,109,110]. Sutada et al. showed that *Ctsk* deletion in osteocytes prevented the increase in the osteocyte lacunar area observed during lactation, as well as the effects of lactation to increase osteoclast numbers and decrease trabecular bone volume, cortical thickness, and mechanical properties [111]. Mutations in *TCIRG1* result in dense and fragile bone, caused by a defect in osteoclasts responsible for bone destruction [112]. Normally expressed CTSK and TCIRG1 are the basis for osteoclasts to perform bone turnover function. The results of the present study showed upregulated CTSK and downregulated TCIRG1 in case group, which may be one cause of glucocorticoid-induced ONFH.

As the number of neutrophils in the case group was 0.51 times that in the control group, we investigated cell proliferation and apoptosis-associated genes. Our results showed that Ccnd1 served as the network hub of multiple pathways and was highly expressed in the case group (Appendix A). Upregulated CCND1 forms a complex with and functions as a regulatory subunit of CDK4 or CDK6, whose activity is required for cell cycle G1/S transition; however, the expression of downstream CDK6 was relatively low in the case group. 

In conclusion, the combined effects of SLE and glucocorticoid therapy on bone marrow involve changes in cell number and biological processes of various cell types, which primarily concentrate on the dysregulation of neutrophil and B cell. Moreover, the cell types surrounding hBMSCs participate in glucocorticoid-induced ONFH, and the changes and interaction of activated or suppressed inflammatory responses, substance metabolism, vascular injury and angiogenesis, cell proliferation and apoptosis, and dysregulated coagulation and fibrinolysis are mainly responsible for glucocorticoid-induced ONFH. However, the regulatory relationship between multiple pathways in the same cell and the cell-to-cell interaction relationship need to be further clarified. 

## Figures and Tables

**Figure 1 biomolecules-13-00171-f001:**
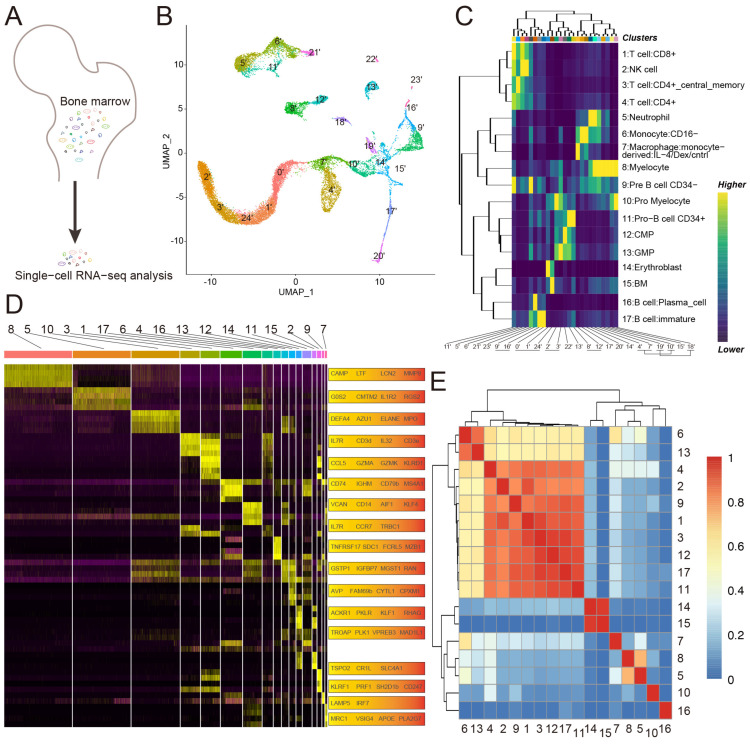
**Single−cell RNA-seq of human bone marrows**. Bone marrow extracted and droplet-based single-cell RNA-seq after rigorous filtration (**A**). Cellular composition of human bone marrow using uniform manifold approximation and projection (**B**). Seventeen cell types formed by 25 clusters (**C**). Seventeen cell types identified by 2 to 4 marker genes (**D**). The correlations of average expression profiles among cell types (**E**).

**Figure 2 biomolecules-13-00171-f002:**
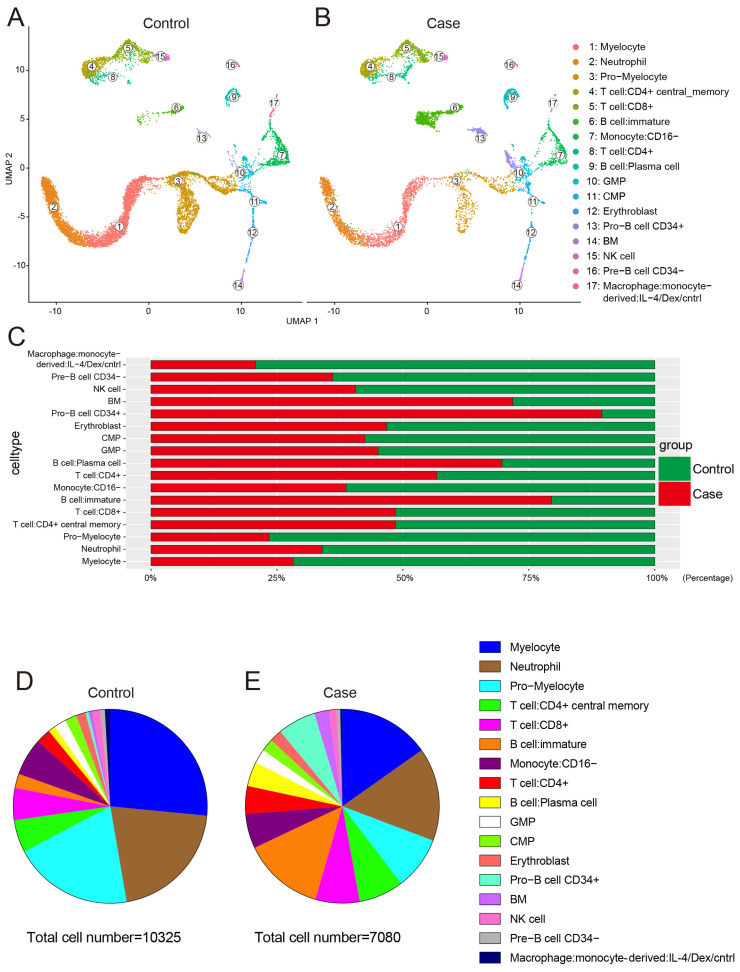
**Profile of cell types and corresponding cell number and proportion**. The distribution of the same 17 cell types in both the case and control groups (**A**,**B**). Comparison of the proportion of the same cell type in the case and control groups (**C**). All cell types and corresponding constituent ratio in the control and case groups (**D**,**E**).

**Figure 3 biomolecules-13-00171-f003:**
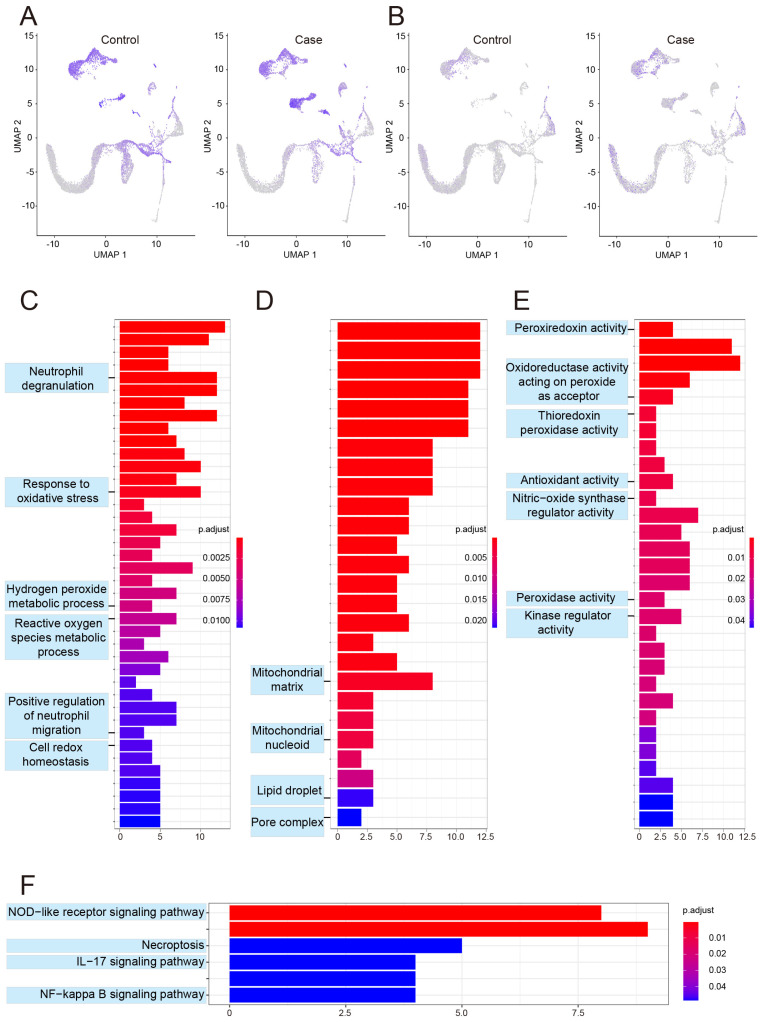
**Distribution and enrichment analysis of femoral head necrosis and glucocorticoid-related genes**. The distribution of ONFH-related genes in cell types (**A**). The distribution of glucocorticoid-related genes in cell types (**B**). The enriched biological processes (**C**), cellular components (**D**), and molecular functions (**E**) of glucocorticoid-related genes. The enriched KEGG signaling pathways of glucocorticoid-related genes (**F**).

**Figure 4 biomolecules-13-00171-f004:**
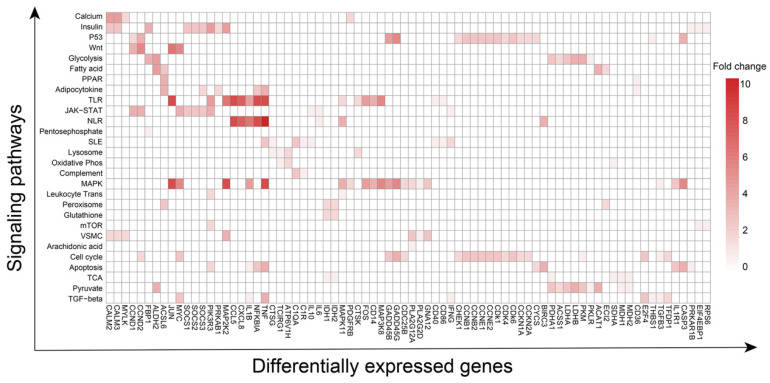
The distribution of 78 differentially expressed genes in 28 signaling pathways in the case and control groups.

**Figure 5 biomolecules-13-00171-f005:**
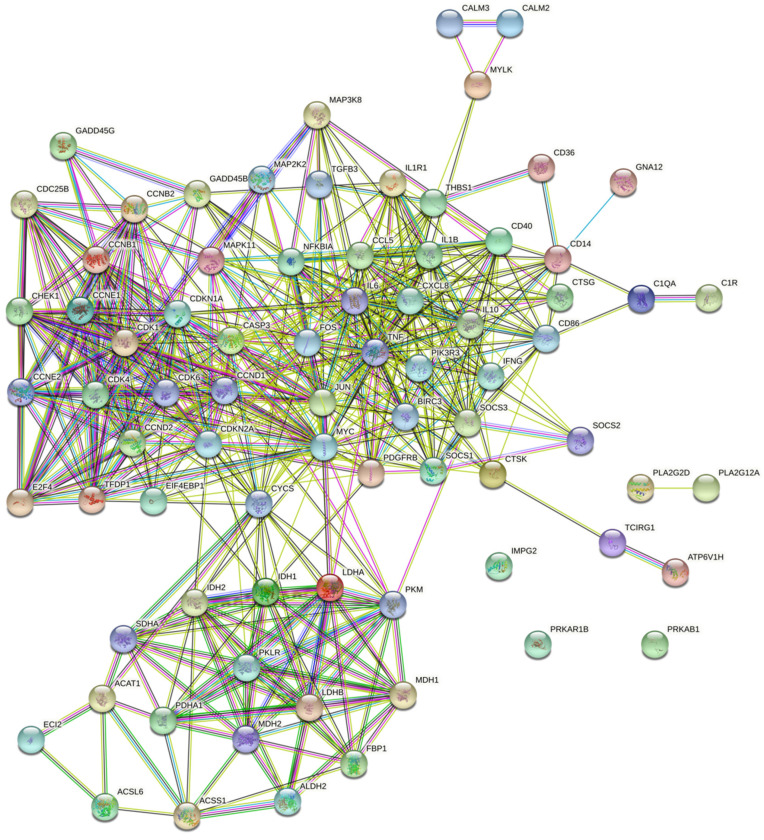
The protein–protein interaction network of 78 proteins encoded by differentially expressed genes involved in 28 signaling pathways.

**Figure 6 biomolecules-13-00171-f006:**
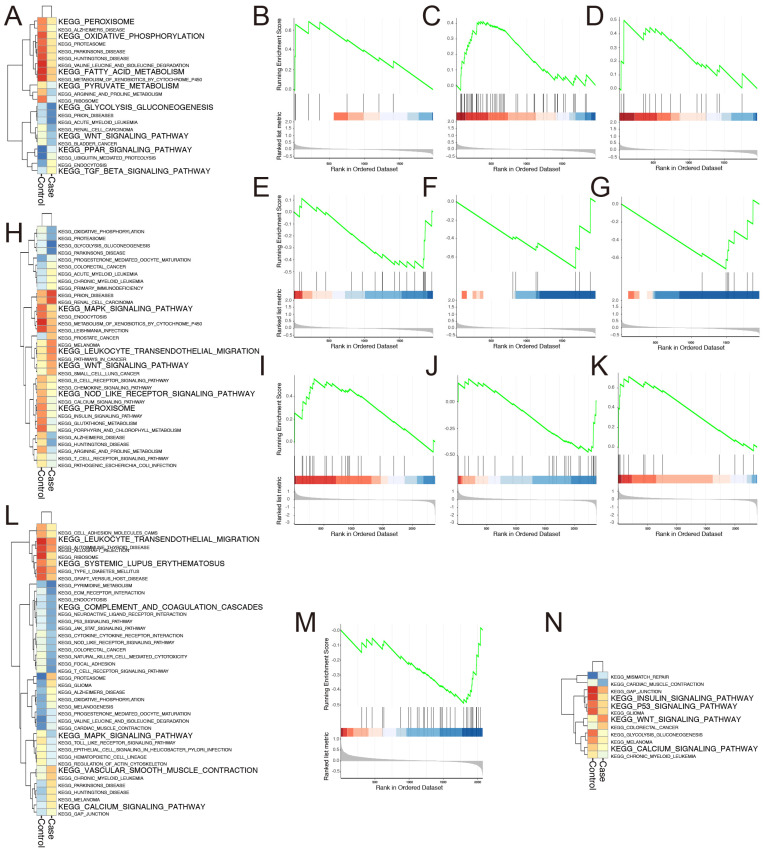
**The results of GSVA and GSEA of erythroblast and monocyte-derived macrophage and immature B cell.** The substance metabolism-associated pathways uncovered by GSVA for erythroblast (**A**). GSEA of erythroblast, showing adipocytokine, P53 signaling pathways, leukocyte trans-endothelial migration, citrate cycle TCA cycle, glutathione metabolism, and cell cycle, respectively (**B**–**G**). The inflammatory response-related pathways analyzed by GSVA for monocyte-derived macrophage (**H**). GSEA of monocyte-derived macrophage, showing JAK-STAT, systemic lupus erythematosus, and complement and coagulation cascades, respectively (**I**–**K**). Vascular injury and bone metabolism correlative pathways of immature B cell (**L**). The repressed lysosome pathway of neutrophil in the case group revealed by GSEA (**M**). Cell proliferation and apoptosis-related signaling pathways demonstrated by GSVA (**N**).

**Figure 7 biomolecules-13-00171-f007:**
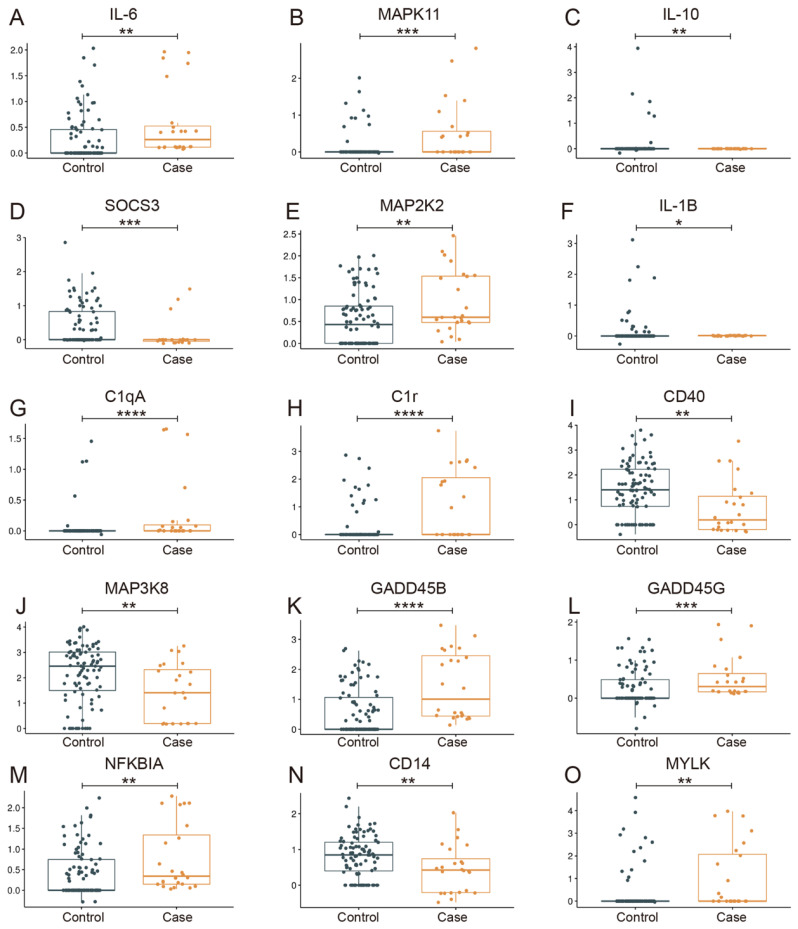
The relative expression level of cytokines of monocyte-derived macrophages in case and control groups. * *p* < 0.05, ** *p* < 0.01, *** *p* < 0.001, **** *p* < 0.0001.

**Table 1 biomolecules-13-00171-t001:** **The affected signaling pathways in various cell types in bone marrow.** PPAR: peroxisome proliferators-activated receptor, NLR: NOD-like receptor, TLR: toll-like receptor, MAPK: mitogen-activated protein kinase.

	Myelocyte	Neutrophil	Pro-Myelocyte	T Cell: CD4+ Central_Memory	T Cell: CD8+	B Cell: Immature	Monocyte: CD16−	T Cell: CD4+	B Cell: Plasma Cell	GMP	CMP	Erythroblast	Pro B Cell: CD34+	BM	NK Cell	PreB Cell CD34−	Macrophage: Monocyte Derived
Lysosome	√	—	—	—	—	√	—	—	—	√	—	—	—	—	—	—	√
Wnt signaling pathway	√	√	√	—	√	—	—	—	—	—	√	√	√	—	√	√	√
Pyruvate metabolism	√	—	—	—	—	—	√	√	—	√	—	√	—	—	—	√	—
Insulin signaling pathway	—	√	—	—	√	—	√	√	—	—	—	—	√	—	—	—	√
P53 signaling pathway	—	√	—	√	—	√	—	—	—	√	√	√	—	—	—	√	—
Glycolysis and gluconeogenesis	—	√	—	—	√	—	√	√	—	√	√	√	√	—	√	—	√
Calcium signaling pathway	—	√	√	—	—	√	—	√	√	—	—	—	—	—	—	√	√
NLR signaling pathway	—	—	√	√	√	√	—	√	√	√	√	—	√	—	√	—	√
MAPK signaling pathway	—	—	√	√	√	√	√	√	√	√	√	—	—	√	—	√	√
Complement and coagulation cascades	—	—	√	√	—	√	—	√	—	√	—	—	—	—	—	√	√
Oxidative phosphorylation	—	—	√	√	—	√	—	—	—	—	—	√	—	—	—	—	√
TLR signaling pathway	—	—	—	√	√	√	—	√	√	√	—	—	√	—	√	—	—
TGF-Beta signaling pathway	—	—	—	√	—	—	—	√	—	—	—	√	—	—	—	√	—
Apoptosis	—	—	—	√	√	—	—	√	—	—	√	—	—	—	—	—	—
Peroxisome	—	—	—	√	—	—	√	—	—	—	—	√	—	—	√	—	√
Systemic lupus erythematosus	—	—	√	—	—	√	—	—	—	—	—	—	—	—	—	—	√
JAK-STAT signaling pathway	—	—	—	—	—	√	—	—	—	—	—	—	√	—	—	—	√
Vascular smooth muscle contraction	—	—	—	—	—	√	—	—	—	—	—	—	—	√	√	√	—
Leukocyte trans-endothelial migration	—	—	—	—	—	√	—	—	—	√	—	√	—	—	—	—	√
PPAR signaling pathway	—	—	—	√	—	—	√	—	—	√	—	√	√	—	—	√	—
Fatty acid metabolism	—	—	—	—	—	—	√	√	√	—	—	√	√	—	—	√	—
Cell cycle	—	—	—	—	—	—	—	√	—	—	√	√	—	—	—	√	—
Glutathione metabolism	—	—	—	—	—	—	—	—	—	—	—	√	—	—	—	√	√
Adipocytokine signaling pathway	—	—	—	√	—	—	—	—	—	—	—	√	√	—	—	—	—
mTOR signaling pathway	—	—	—	—	√	√	—	—	—	—	—	—	—	—	—	√	—
NOTCH signaling pathway	—	—	—	—	—	—	√	—	√	—	—	—	—	—	—	—	√
Citrate cycle TCA cycle	—	—	—	—	—	—	—	—	—	—	—	√	—	—	—	—	—
Pentose phosphate pathway	—	—	—	—	—	—	—	—	—	—	—	—	—	—	—	—	√

## Data Availability

The single-cell RNA-seq data, quality control information, and cluster information are available from the corresponding authors on reasonable request.

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
