# Peer review of "Accumulation of Fat Not Responsible for Femoral Head Necrosis, Revealed by Single-Cell RNA Sequencing: A Preliminary Study"

_biomolecules, 2023, doi:10.3390/biom13010171_

Round 1

Reviewer 1 Report

the authors applied RNA seq on single cells on bone marrow specimens. to my knowledge no one investigate the bone marrow of an "healthy" donor, of course,  it is also not feasible and ethic collect bone marroe from healthy subject.  

I found the manuscript original, useful to implement the knowledge about biology of bone marrow and on development of ONFH, focusing on the effect exerted by glucocorticosteroids. 

Morover it  was difficult for the reader understand the manuscirpt, for the amount of data and pathway reported. I suggest to the authors, if it is possibile to summerize the results. 

one comments: the authors need to bette clarify the "control" samples. in introduction they said that use two pateints with SLE and two patients undergone to hip arthroplasty, leading the reader to think that they are using patients not affected by ONFH, but with a trauma event. On contrary in material and section all patients involved are affected by ONFH.  if feasible, it could be more useful compare data with "healthy" donours. 

Author Response

Response to Reviewer’s comments

Reviewer #1

The authors applied RNA seq on single cells on bone marrow specimens. To my knowledge no one investigate the bone marrow of a "healthy" donor, of course, it is also not feasible and ethic collect bone marrow from healthy subject. I found the manuscript original, useful to implement the knowledge about biology of bone marrow and on development of ONFH, focusing on the effect exerted by glucocorticosteroids. Moreover, it was difficult for the reader understand the manuscript, for the amount of data and pathway reported. I suggest to the authors, if it is possible to summarize the results.

Author reply: We highly appreciate the reviewer for this positive comment. We have rephrased the Section of Results and Discussion as possible. The revised content was marked with blue words in Page 4, 9, 14,16, 18, 20, and 21.

  1. the authors need to better clarify the "control" samples. in introduction they said that use two patients with SLE and two patients undergone to hip arthroplasty, leading the reader to think that they are using patients not affected by ONFH, but with a trauma event. On contrary in material and section all patients involved are affected by ONFH. If feasible, it could be more useful compare data with "healthy" donors.

Author reply: Thanks for your considerate reminding. As mentioned in the abstract, the case group consisted of the patients diagnosed with ONFH for systemic lupus erythematosus (SLE) treated with glucocorticoids and the control group consisted of the patients with femoral neck fracture not affected by ONFH. The corresponding contents in the manuscript have been revised as below:

Page 2 Paragraph 3, Section of Introduction: “In this study, bone marrow samples from the proximal femur were extracted from two patients with ONFH for SLE and two with femoral neck fracture during hip arthroplasty.”

Page 3 Paragraph 2, Section of Materials and Methods: “Bone marrow of the proximal femur was extracted from two male participants (39 and 71 years old) diagnosed with femoral neck fracture and two female participants (19 and 32 years old) diagnosed with ONFH after glucocorticoid therapy for SLE during hip arthroplasty.”

Reviewer 2 Report

The manuscript is an interesting study, using high-class analytical methods. It presents very interesting results. However, I have some remarks:
1. There is no information in the Materials and Methods chapter about the approval of the ethics committee
2. Very small number of patients - 2 patients in each group with large differences in age! In addition, there is no information on how long glucocorticoid therapy has been used in SLE patients. In my opinion, the research should be treated as a preliminary study. If the manuscript will be published, this should be included in the title.
3. In the Results chapter, Authors often refer to the results and statements of other authors. They also indicate probable mechanisms. These fragments should be remodeled, i.e. transferred to the discussion. Due to the large number of results and their complexity, the authors should refer to the documentation more often. Sometimes there are no references - e.g. Enolase 1 (ENO1) is downregulated in case group....

4. Table 1 must be corrected in its edition

5. In my opinion, some figures are hard to read, e.g. Fig. 6 A, H, L, N

Author Response

Response to Reviewer’s comments

Reviewer #2

The manuscript is an interesting study, using high-class analytical methods. It presents very interesting results.

Author reply: We highly appreciate the reviewer for this positive comment.

  1. There is no information in the Materials and Methods chapter about the approval of the ethics committee.

Author reply: Thank you for this valuable comment. The information about the ethics approval statement had been added as below:

Page 3, Paragraph 3, Section Materials and Methods: Obtaining bone marrow samples of femoral head necrosis, adding the paragraph “This study was approved by the Ethics Committee of Peking Union Medical College Hospital (No. JS-2447) and was conducted in accordance with the International Conference on Harmonization Good Clinical Practice guideline, the Declaration of Helsinki, and applicable regulatory requirements. All patients provided written informed consent prior to participation in the study.”

  1. Very small number of patients - 2 patients in each group with large differences in age! In addition, there is no information on how long glucocorticoid therapy has been used in SLE patients. In my opinion, the research should be treated as a preliminary study. If the manuscript will be published, this should be included in the title.

Author reply: Thank you very much for this suggestion. The present study does not focus on the aging related genes and the osteonecrosis of femoral head is also not an age-related disease. Additionally, we have been continuing to collect bone marrow samples for subsequent research and the relevant research results will be further shared. The information about the glucocorticoid therapy had been added in the Section Materials and Methods: Obtaining bone marrow samples of femoral head necrosis as below:

Page 3, Paragraph 2, adding the sentence “The two SLE patients were diagnosed with ONFH by magnetic resonance imaging 1 year and 5 years respectively after methylprednisolone therapy, and the cumulative dosage of methylprednisolone was more than 4g and 15g, respectively.”

As suggested, the title “Accumulation of fat not responsible for femoral head necrosis: revealed by single-cell RNA sequencing” has been replaced with “Accumulation of fat not responsible for femoral head necrosis: revealed by single-cell RNA sequencing: A preliminary study”.

  1. In the Results chapter, Authors often refer to the results and statements of other authors. They also indicate probable mechanisms. These fragments should be remodeled, i.e. transferred to the discussion. Due to the large number of results and their complexity, the authors should refer to the documentation more often. Sometimes there are no references - e.g. Enolase 1 (ENO1) is downregulated in case group....

Author reply: Thanks for your suggestion. Some results of other authors had been transferred to the Section of Discussion, such as below:

Page 20, the last Paragraph rephrased as “It had been reported that an immature B cell population can serve as a marker for monitoring tumor angiogenesis and anti-angiogenesis therapy in mouse models (Fagiani et al., 2015). Additionally, the dysregulation of immature B cell is associated with bone loss (Titanji et al., 2020). For immature B cell, activated calcium and vascular smooth muscle contraction and suppressed MAPK pathways influence vascular function and angiogenesis. Moreover, upregulated MYLK, a common gene of calcium and vascular smooth muscle contraction pathways, triggers endothelial contraction. Myc is downreg-ulated, which can activate the transcription of growth-related genes and bind to the VEGFA promoter to facilitate VEGFA production and subsequent sprouting angiogenesis (Shi et al., 2014) and platelet-derived growth factor receptor beta (PDGFRB), which is essential for normal blood vessel development and repair of vascular injury sites (Hosaka et al., 2020). Additionally, complement and coagulation cascade pathways activated by immune complexes were suppressed in patients with SLE treated by glucocorticoids, indicating that activated complement does not damage the blood vessels, which may be responsible for the immunosuppressive effects of glucocorticoids.”

The relevant references had been cited in the corresponding parts such as below:

Page 18, Paragraph 2-3: “Increased GTSE1 binds to the tumor suppressor protein p53 to prevent apoptosis(Monte et al., 2003; Monte et al., 2004). Besides GTSE1, other genes are also known to participate in cell apoptosis: For instance, (1) in-creased VDAC1 may participate in the formation of the permeability transition pore complex (PTPC), which is responsible for the release of mitochondrial products that trigger apoptosis (Li et al., 2014; Verrier et al., 2003); (2) decreased EIF4EBP1 and RRM2 facilitate cell proliferation(Gui et al., 2021; T. Li et al., 2022); (3) increased CCND1 forms a complex with, and functions as a regulatory subunit of CDK4 or CDK6, whose activity is required for cell cycle G1/S transition (CDK6 expression was relatively low in the case group) (Lu et al., 2022; Yang et al., 2022); and (4) low expression of CHEK1 and CDKN1A serve to inhibit cell proliferation (Jin et al., 2006; Ou et al., 2005; Shieh et al., 2000).

Enolase 1 (ENO1) is downregulated in case group (SI Fig. 9A), which promote the cell proliferation (Li et al., 2021; Y. Li et al., 2022). Besides ENO1, thrombospondin 1 (THBS1), the ligand for CD36, which has antiangiogenic properties and cell-to-cell/matrix inter-actions(Farberov & Meidan, 2014; Kaur et al., 2021), was higher in the case group (SI Fig. 9B).”

Adding SI Figure 9: To provide the data of expression level of ENO1 and THBS1 in the case group compared with the control group, we have added one picture in the Section of Supporting Information as SI Figure 9.

SI Fig. 9. The relative expression level of ENO1 and THBS1 in the case and control groups.

  1. Table 1 must be corrected in its edition.

Author reply: Thank you for this comment. This is a good suggestion to improve the manuscript. According to this comment, Table 1 had been corrected in Page 11.

  1. In my opinion, some figures are hard to read, e.g. Fig. 6 A, H, L, N.

Author reply: We thank the Reviewer for this comment. We have read and revised the content of Figures in the text and the content of the Fig.6 is explained in detail in the text as below:

Page 14, Paragraph 2: “The results of GSVA and GSEA indicated that the substances metabolism of eryth-roblasts changed the most. GSVA revealed that the regulated pathways in the case group included down-regulated peroxisome, oxidative phosphorylation, fatty acid metabolism, pyruvate metabolism, glycolysis gluconeogenesis, Wnt pathways and up-regulated PPAR and TGF-beta pathways (Fig. 6A). Additionally, GSEA demonstrated up-regulated adipocytokine (Fig. 6B), P53 signaling pathways (Fig. 6C), leukocyte trans-endothelial migration (Fig. 6D) as well as down-regulated citrate cycle TCA cycle (Fig. 6E), gluta-thione metabolism (Fig. 6F), cell cycle (Fig. 6G).”

Page 16, Paragraph 3: “The results of GSVA and GSEA indicated that the inflammation-related pathways of monocyte-derived macrophage changed the most. The most significantly affected in-flammation-related pathways in the case group were down-regulated MAPK, NLR, peroxisome, and systemic lupus erythematosus, and up-regulated leukocyte trans-endothelial migration, Wnt, JAK-STAT, and complement and coagulation cascades pathways (Fig. 6H–K), all of which are associated with SLE, inflammatory response, blood vessel development, and glucocorticoid therapy.”

Page 18, Paragraph 1: “The MAPK pathway was found to be repressed, and the vascular smooth muscle contraction pathway was activated in the case group (Fig. 6L)”

Page 18, Paragraph 2: “GSVA demonstrated activation of the Wnt pathway, and inhibition of insulin, P53, cal-cium pathways, and the glycolysis gluconeogenesis pathway in case group (Fig. 6N)”

Round 2

Reviewer 1 Report

The manuscript was improuved after revision. I finally suggest to the authors to justify the decision to enrolled only 2 patients, since that in biology field it's recomend to collect more specimens, at leas 3 for group.   

Author Response

Response to Reviewer’s comments-Round 2

Reviewer #1

The manuscript was improved after revision. I finally suggest to the authors to justify the decision to enrolled only 2 patients, since that in biology field it's recommend to collect more specimens, at least 3 for group.

Author reply: Thanks for this positive comment. It is a meaningful suggestion to increase the number of samples. Three biological repeats are generally required for bulk RNA sequencing. While one cell can be regarded as a duplication for single cell RNA sequencing. In our research, 7080 and 10325 individual cells in the case and control groups were analyzed, respectively. For instance, Reuben Moncada et al.reported the results of single cell RNA sequencing using fresh primary pancreatic ductal adenocarcinoma from two patients (Nat Biotechnol. 2020, 38(3):333-342), and single cell RNA sequencing was employed for two tumor samples from mouse (J Immunother Cancer. 2022, 10(5):e004691).